# Chenodeoxycholic Acid-Amikacin Combination Enhances Eradication of *Staphylococcus aureus*

Kaiyu Cui,[a] Weifeng Yang,[a] Zhiyuan Liu,[b] Guijian Liu,[b] Dongying Li,[a] Yanan Sun,[a] Gaiying He,[a] Shuhua Ma,[a] Yu Cao,[c] Xuefan Jiang,[d] (ORCID) Sylvie Chevalier,[e] Pierre Cornelis,[e] (ORCID) Qing Wei,[f] Yi Wang[a]

[a]Experimental Research Center, China Academy of Chinese Medical Sciences, Beijing, China

[b]Guang'anmen Hospital, China Academy of Chinese Medical Sciences, Beijing, China

[c]Xiyuan Hospital, China Academy of Chinese Medical Sciences, Beijing, China

[d]Beijing University of Chinese Medicine, Beijing, China

[e]Normandy University, University of Rouen Normandy, Laboratory of Microbiology Signals and Microenvironment, Evreux, France

[f]Nanchang Institute of Technology, Nanchang, Jiangxi, China

Kaiyu Cui and Weifeng Yang contributed equally to this work. Author order was determined on the basis of contribution.

**ABSTRACT** The rise of antibiotic resistance and dearth of novel antibiotics have posed a serious health crisis worldwide. In this study, we screened a combination of antibiotics and nonantibiotics providing a viable strategy to solve this issue by broadening the antimicrobial spectrum. We found that chenodeoxycholic acid (CDCA), a cholic acid derivative of the traditional Chinese medicine (TCM) Tanreqing (TRQ), synergizes with amikacin against *Staphylococcus aureus in vitro*, and this synergistic killing was effective against diverse methicillin-resistant *S. aureus* (MRSA) variants, including small-colony variants (SCVs), biofilm strains, and persisters. The CDCA-amikacin combination protects a mouse model from *S. aureus* infections. Mechanistically, CDCA increases the uptake of aminoglycosides in a proton motive force-dependent manner by dissipating the chemical potential and potentiates reactive oxygen species (ROS) generation by inhibiting superoxide dismutase activity. This work highlights the potential use of TCM components in treating *S. aureus*-associated infections and extend the use of aminoglycosides in eradicating Gram-positive pathogens.

**IMPORTANCE** Multidrug resistance (MDR) is spreading globally with increasing speed. The search for new antibiotics is one of the key strategies in the fight against MDR. Antibiotic resistance breakers that may or may not have direct antibacterial action and can either be coadministered or conjugated with other antibiotics are being studied. To better expand the antibacterial spectrum of certain antibiotics, we identified one component from a traditional Chinese medicine, Tanreqing (TRQ), that increased the activity of aminoglycosides. We found that this so-called agent, chenodeoxycholic acid (CDCA), sensitizes *Staphylococcus aureus* to aminoglycoside killing and protects a mouse model from *S. aureus* infections. CDCA increases the uptake of aminoglycosides in a proton motive force-dependent manner by dissipating the chemical potential and potentiates ROS generation by inhibiting superoxide dismutase activity in *S. aureus*. Our work highlights the potential use of TCM or its effective components, such as CDCA, in treating antibiotic resistance-associated infections.

**KEYWORDS** antibiotic resistance, *Staphylococcus aureus*, traditional Chinese medicine, chenodeoxycholic acid, dual mechanism

Gram-positive (G+) *Staphylococcus aureus* infection has always posed an important threat to life in human health care (1, 2). A wide variety of diseases are associated with *S. aureus*, such as skin infections, food poisoning, joint infection, and septicemia

Address correspondence to Qing Wei, vubwqing@hotmail.com, or Yi Wang, prof.wangyi@foxmail.com.

The authors declare a conflict of interest. Experimental Research Center, China Academy of Chinese Medical Sciences, January 18, 2022. China Patent application 202210055969.3.

10.1128/spectrum.02430-22 **1**

(3–5). The underlying mechanisms of action are based on intrinsic multifunctional toxins and the emergence of multidrug resistance (MDR) (6). MDR in *S. aureus* has gained increasing attention, and MDR *S. aureus* has been prioritized on a list of bacteria by the World Health Organization to ensure that it responds to this urgent health need (7). Particularly, the rise of methicillin-resistant *S. aureus* (MRSA) and vancomycin (VAN)-resistant strains has led to unprecedented failures of antibiotic treatments (8, 9). Given the growing MDR and a paucity of antibacterial products, there is an urgent need for alternative treatments to conquer this battle against bacterial infections.

Combinations of antibiotics have been proposed and used in clinical medicine for decades to broaden the antimicrobial spectrum and induce synergistic effects (10–12). A classic example is the treatment of *Mycobacterium tuberculosis* infections by multiple drugs, including rifampin, isoniazid, and many others (11–13). Alternatively, combinations of antibiotics and nonantibiotics are receiving extensive attention, among which $\beta$-lactam antibiotic–$\beta$-lactamase inhibitor pairs are an excellent example (10, 12). The combinatory use of these antibiotics could slow down the development of antibiotic resistance and extend the antibacterial spectrum of several antibiotics, including changes from targeting Gram-negative (G$^-$) or G$^+$ bacteria to targeting both types of infections.

Previously, we demonstrated that injection of the traditional Chinese medicine (TCM) Tanreqing (TRQ) could effectively repress the regulation of virulence in *Pseudomonas aeruginosa* (14) and suppress the biofilm formation (15) and virulence (16) of *S. aureus*. TRQ is a Chinese herbal preparation extracted from *Scutellariae radix* (Huang Qin), *Lonicerae flos* (Jin Yin Hua), *Forsythiae fructus* (Lian Qiao), Ursi Fel (Xiong Dan), and *Naemorhedi cornu* (Shan Yang Jiao) and has been used in China for a long time (15, 17). It was postulated that TRQ injection could eliminate fever, detoxify, and remove phlegm and could be used as a treatment for respiratory tract infections, pneumonia, and chronic obstructive pulmonary disease (18). However, the chemical basis of TRQ is not yet well understood with respect to its antimicrobial activity against both G$^-$ and G$^+$ pathogens.

Cholic acid (CA) is known for its amphiphilic nature, which mimics naturally occurring antimicrobial peptides (19, 20). Numerous CA derivatives have been extensively studied with respect to their involvement in immunity and aging (21, 22). Specifically, one of the CA derivatives, isoallolithocholic acid (isoalloLCA), acts as a T cell regulator in mice (21) and more recently demonstrated striking antimicrobial effects against G$^+$ pathogens such as *Clostridioides difficile* and *Enterococcus faecium* (22). In addition, the antimicrobial nature of CA was extensively investigated (23–25). Besides, CA and its derivatives have been approved by health authorities such as the European Medicines Agency, Food and Drug Administration, and National Medical Products Administration to protect against inborn errors of primary bile acid synthesis (26). However, there is limited information on the combinatory application of CA derivatives with traditional antibiotics against clinical pathogens.

Amikacin (AMK) belongs to aminoglycoside family antibiotics mainly used for G$^-$ infections and has led to tremendous success in critical care units (27–29). To further extend its shelf life and antibacterial spectrum, combinations with other antibiotics or nonantibiotics could be investigated. In this study, we aimed to unravel the mode of action of TRQ against *S. aureus* infections with combinatory usage of antibiotics and nonantibiotic agents. We found that one of the effective components of Ursi Fel in TRQ, named chenodeoxycholic acid (CDCA), could dissipate the chemical potential ($\Delta$pH) of proton motive force (PMF) and sensitize *S. aureus* to aminoglycoside killing *in vitro*. In addition, this synergistic killing was demonstrated to be effective in diverse MRSA variants. Finally, we demonstrate that CDCA could efficiently protect a mouse infection model from *S. aureus* infections *in vivo*. This work highlights the potential use of TCM or its effective components in treating antibiotic resistance-associated infections and provides insights into mode of action of these antimicrobial agents.

## RESULTS

**CA derivatives inhibit *S. aureus* growth.** Previously, we showed that the traditional Chinese medicine TRQ could inhibit the growth and virulence of *S. aureus* (16, 30). To

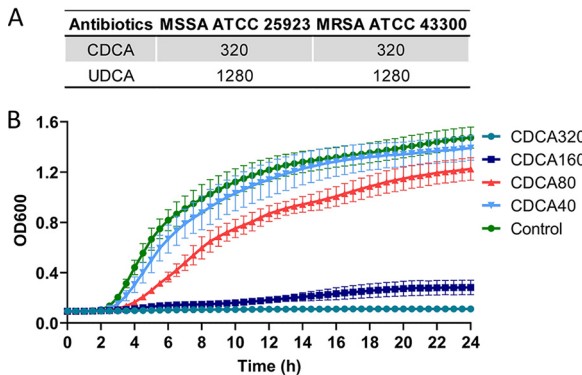

**FIG 1** CA derivatives inhibit the growth of *S. aureus*. (A) MICs of CDCA and UDCA on MSSA strain ATCC 25923 and MRSA strain ATCC 43300, respectively; (B) growth curves of MRSA strain ATCC 43300 at different concentrations of CDCA. Cells were grown in triplicate at 37°C with shaking; the curves represent average values from triplicates. The error bar represents the mean ± standard error of the mean (SEM).

better understand the mode of action of the TRQ formula, a chemical screening analysis was performed showing that one of the TRQ components, chenodeoxycholic acid (CDCA), demonstrated a potent inhibitory effect on the growth of methicillin-sensitive *S. aureus* (MSSA) and MRSA with a MIC of 320 $\mu$g/mL (Fig. 1A). In addition, we showed that another CA derivative, ursodeoxycholic acid (UDCA), exhibited antistaphylococcal activity against both MSSA and MRSA strains with a relatively higher MIC value of 1,280 $\mu$g/mL (Fig. 1A). Growth curve analysis showed that CDCA at the MIC could significantly repress the viability of MRSA (160 $\mu$g/mL and 320 $\mu$g/mL), a low level of CDCA (80 $\mu$g/mL) exhibited a subinhibitory effect on the growth of MRSA, and CDCA at 40 $\mu$g/mL completely lost growth arrest activity on MRSA (Fig. 1B).

To further expand the antistaphylococcal spectrum of CDCA, we tested the MIC of CDCA in 24 clinical MRSA strains and found the same MIC results in Fig. 1A. In addition, fusidic acid (FA), a structural analog of CDCA and a steroid antibiotic isolated from *Fusidium coccineum* (31), demonstrated significant antistaphylococcal activity against 17 clinical MRSA strains and was ineffective against seven MRSA strains, indicating a distinct anti-*Staphylococcus* mechanism compared to CDCA (see Table S1 in the supplemental material).

Collectively, we showed that CDCA and its analogs could inhibit the viability of *S. aureus*.

**CDCA sensitizes *S. aureus* to aminoglycoside killing.** The combination of antibiotics and nonantibiotics attracted our attention with a focus on the use of CDCA as an adjunct to traditional antibiotics. Besides, the dual therapy could reduce the side effects of individual agents (11). Therefore, we hypothesized that CDCA might function as a synergistic agent when used in combination with antibiotics. Previously, we have shown that TRQ combined with vancomycin (VAN) or linezolid (LZD) exhibited synergy against MRSA (30). In this study, we selected aminoglycoside class antibiotics to investigate their activity against G$^+$ pathogens. As shown in Fig. 2A and B, a lower dose of CDCA significantly increased the anti-*S. aureus* effect of several aminoglycosides, including amikacin (AMK), etimicin, gentamicin, kanamycin, and tobramycin, in both MSSA and MRSA strains. Based on combinatory efficacy, we focus on amikacin, a potent aminoglycoside against *S. aureus*, and determined that MRSA strain ATCC 43300 showed a MIC value of 64 $\mu$g/mL. Specifically, 1/8 MIC of CDCA (40 $\mu$g/mL) and 1/8 amikacin effectively repressed the growth of MSSA (Fig. 2A) and MRSA (Fig. 2B). Meanwhile, the fractional inhibitory concentration index (FICI) for CDCA with different aminoglycosides was 0.25 (Fig. 2C) for both MSSA and MRSA strains, indicating that CDCA is a powerful synergistic agent to reverse resistance to aminoglycosides in *S. aureus* strains. Consistent with this, CDCA and amikacin showed the same effect in 24 clinical MRSA strains (Fig. S1).

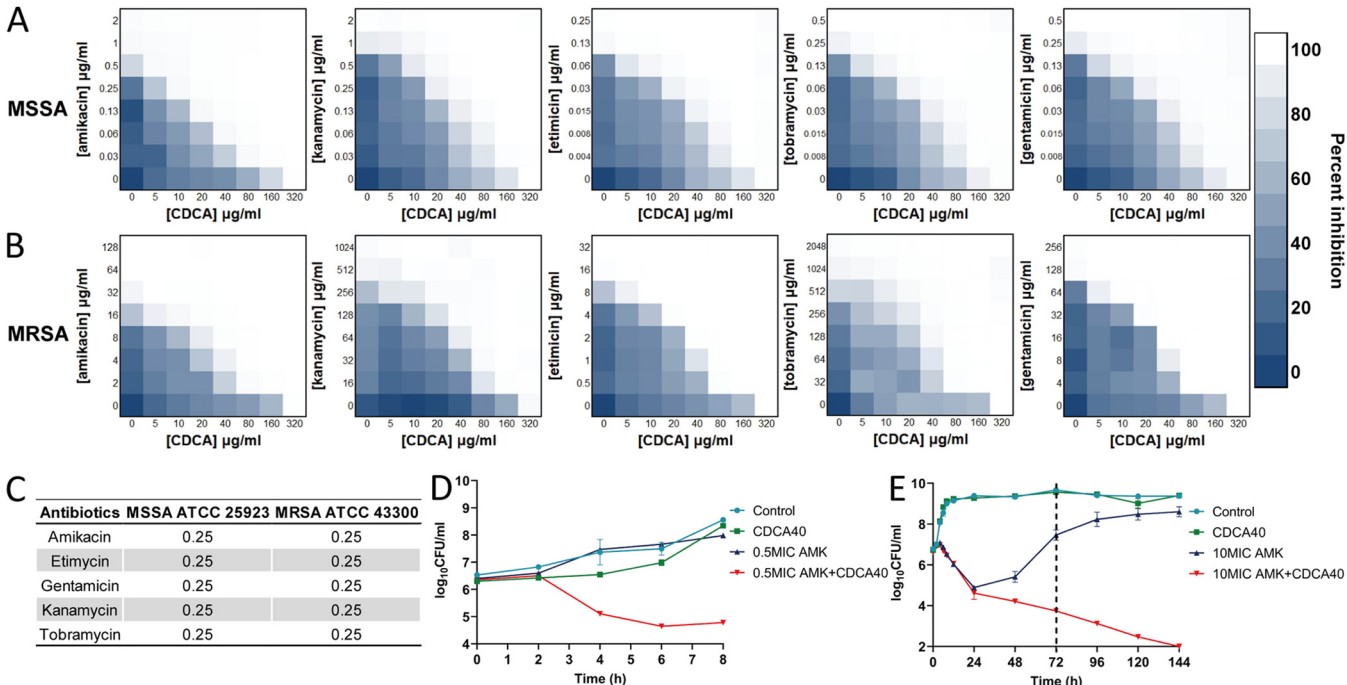

**FIG 2** CDCA sensitizes *S. aureus* to aminoglycoside killing. (A) Synergism between CDCA and aminoglycosides evaluated against MSSA strain ATCC 25923; (B) synergism between CDCA and aminoglycosides evaluated against MRSA strain ATCC 43300; (C) FICI of CDCA in combination with aminoglycoside-treated MSSA strain ATCC 25923 and MRSA strain ATCC 43300. (D) MSSA ATCC 25923 was grown to mid-exponential phase and challenged with 0.5× MIC (0.5 $\mu$g/mL) of amikacin with or without 40 $\mu$g/mL CDCA (CDCA40). The survivors were enumerated at the indicated time points. The error bar represents the mean ± SEM. (E) MRSA ATCC 43300 was grown in the mid-exponential phase and challenged with 10× MIC (640 $\mu$g/mL) of amikacin with or without 40 $\mu$g/mL CDCA. The survivors were enumerated at the indicated time points. The error bar represents the mean ± SEM.

In addition, a bacterial killing assay using a combination of CDCA (1/8× MIC) and amikacin (0.5× MIC for MSSA and 10× MIC for MRSA) demonstrated that the drug combination (which we name CDCAA here) could significantly enhance the inhibitory effect of amikacin on the growth of *S. aureus* strains in a time-dependent manner (Fig. 2D and E). It is noteworthy that CDCA could completely eradicate MRSA in the presence of amikacin (10× MIC), while the growth of MRSA with amikacin alone (10× MIC) was restored to the control level (>10⁸ CFU/mL). However, individually neither CDCA nor amikacin was effective against the growth of *S. aureus*. Similarly, the CDCA analog UDCA demonstrated FICIs of 0.3125 for amikacin and 0.375 for the rest of the aminoglycosides tested (Table S2).

In conclusion, we have shown that the CDCAA strategy is a newly identified approach against *S. aureus* and that CDCA could sensitize MRSA strains to aminoglycoside killing.

**CDCA represses aminoglycoside resistance development.** As we have proven that CDCAA could efficiently eradicate MSSA and MRSA, we argued whether the development of aminoglycoside resistance would occur under the pressure of CDCA alone or in combination with antibiotics. First, we serially passaged three independent MSSA cultures for 30 days with a sub-MIC level of CDCA and found an undetectable resistance phenotype (Fig. 3A). On the contrary, serial passage with penicillin, an antibiotic used for the control of *S. aureus* infections (15), gradually generated resistant strains with a 128-fold MIC of the original control strains (parental strains).

Second, to identify the potential development of resistant mutants of *S. aureus*, we again serially passaged MSSA using three independent treatments with CDCAA (amikacin plus 40 $\mu$g/mL CDCA [AMK+CDCA40]) and found that resistant mutants were generated after 7 days, but the MIC did not increase significantly until after 30 days, to 8 $\mu$g/mL for amikacin. Strikingly, three independent treatments with amikacin alone yielded mutants after 2 days and greatly enhanced the development of resistance that was 128-fold higher than that of the original MSSA strains (Fig. 3B). Moreover, when

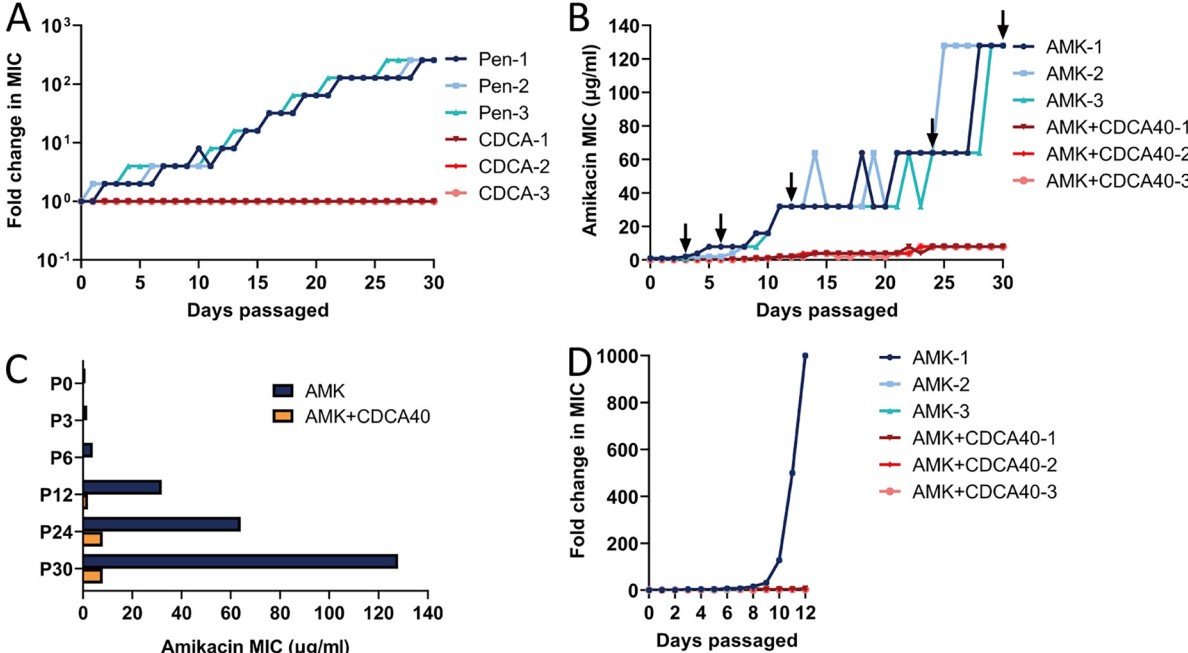

**FIG 3** CDCA represses aminoglycoside resistance development. (A) Evolution of resistance to penicillin (blue) or CDCA (red) in MSSA strain ATCC 25923 after 30 days of passaging in liquid MH medium. Six independent lineages of MSSA strain ATCC 25923 were passaged every 24 h. (B) Evolution of resistance to amikacin without CDCA (AMK-1~3) or with 40 μg/mL CDCA (AMK+CDCA40-1~3) in six independent lineages of MSSA strain ATCC 25923 after 30 days of passage in liquid MH medium. The arrows indicate the sampling time for strains used in panel C. (C) Minimum amikacin concentration necessary to inhibit the growth of resistant isolates from passaged MSSA strain ATCC 25923; (D) comparison of the change in amikacin MIC with or without CDCA for 12 days on MRSA strain ATCC 43300.

we treated the resistant mutant strains that arose during the passage with amikacin alone, we found that CDCAA could restore amikacin susceptibility of these strains (Fig. 3C). Parallel to MSSA strains, we also investigated MRSA strains with a focus on resistance development in the presence of CDCAA. Of note, we observed a sharp increase in amikacin resistance when treated with antibiotics alone (>1,000-fold) and the CDCAA combination gave rise to a slower increase in the development of resistance after 12 days (4-fold) (Fig. 3D).

In conclusion, we showed that CDCA (and CDCAA) could repress the development of aminoglycoside resistance. More importantly, we have demonstrated that CDCA sensitized both MSSA and MRSA strains to aminoglycoside killing.

**CDCA potentiates aminoglycoside activity against MRSA variants.** As we have already shown that CDCAA could significantly enhance the killing effect of amikacin against *S. aureus* and MRSA strains, we asked whether CDCAA was applicable to clinically important conditions, such as *S. aureus* biofilm formation. First, we evaluated the effect of CDCAA on preformed biofilms of *S. aureus*, a phenotype involved in chronic infections, and observed that CDCAA significantly reduced the biomass of biofilms (Fig. 4A). As a control, the medium control and dimethyl sulfoxide (DMSO) solvent alone had no inhibitory effect on the preformed biofilms of *S. aureus*. Bacterial adherence on the surface of plastic was observed by using CLSM (Fig. 4B and Fig. S2). All controls, including treatments with amikacin alone, showed a typical biofilm phenotype, although it was observed that a higher concentration (16× MIC) led to more pronounced bacterial cell death. In contrast, upon addition of CDCA, CDCAA treatment exhibited much less colonization of *S. aureus* cells in a CDCA concentration-dependent manner.

The small-colony variant of *S. aureus* (SaSCV) is known to have antibiotic resistance, is recalcitrant to eradicate, and is frequently isolated from clinical settings (32). To examine whether CDCAA could sensitize SaSCV to antibiotic killing, we generated SaSCVs by selecting auxotrophic mutants by subculturing the MRSA strain on Columbia sheep

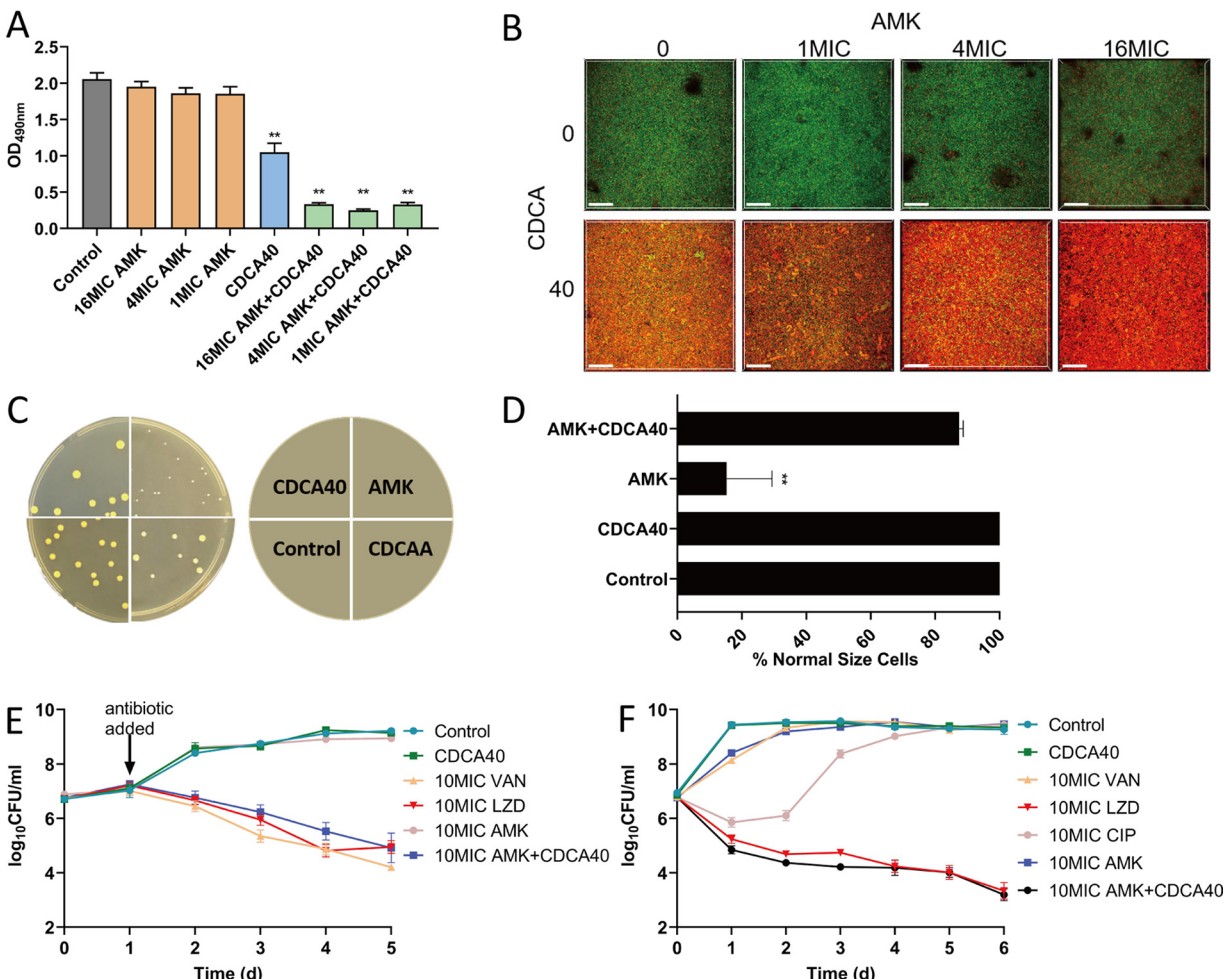

**FIG 4** CDCA potentiates aminoglycoside activity against MRSA variants. (A) Effect of amikacin combined with CDCA on the mature MRSA biofilm. The error bar represents the mean ± SEM. Significance was tested by one-way ANOVA (**, $P < 0.001$). (B) Representative $z$ axis (25 $\mu$m) overlay image of MRSA strain ATCC 43300 biofilms (triplicates) treated with amikacin alone (top) or in combination with CDCA (bottom). The biofilms were stained with the BacLight bacterial viability kit, which visualized dead cells in red with propidium iodide and living cells in green with SYTO 9. Biofilms were evaluated in at least three independent experiments. The scale bar is 100 $\mu$m. (C) Representative images of cell morphology collected in MRSA strain ATCC 43300 treated with different combinations of compounds. (D) Normal-size cells as a proportion of all cells in MRSA strain ATCC 43300 treated with different compounds. The error bar represents the mean ± SEM. Significance was tested by one-way ANOVA (**, $P < 0.0001$). (E) The *S. aureus* SCV strain was grown for 24 h and challenged with 10× MIC (640 $\mu$g/mL) of amikacin with or without 40 $\mu$g/mL CDCA; 10× MIC of LZD (10 $\mu$g/mL), and 10× MIC of VAN (10 $\mu$g/mL) were used as controls. The CFU enumeration was carried out at indicated time points. The error bar represents the mean ± SEM. (F) Amikacin in combination with CDCA kills persisters surviving ciprofloxacin treatment. The *S. aureus* persister strain was grown for 24 h and challenged with 10× MIC (640 $\mu$g/mL) of amikacin with or without 40 $\mu$g/mL CDCA; 10× MIC of LZD (10 $\mu$g/mL), 10× MIC of VAN (10 $\mu$g/mL), and 10× MIC of CIP (2.5 $\mu$g/mL) were used as controls.

blood agar plates (Fig. S3). As can be seen in Fig. 4C and D, we observed that CDCA (or CDCAA) could markedly repress the emergence of SaSCV. As a control, amikacin alone did not exhibit an inhibitory effect on SaSCV growth, suggesting that CDCA treatment could curtail the development of SaSCV. Killing curve analysis showed that CDCA resensitized SaSCV to amikacin (2 logs) (Fig. 4E). As positive controls, vancomycin (VAN) and linezolid (LZD), two antibiotics used for SaSCV sterilization, showed an inhibitory effect on SaSCV similar to that of CDCAA treatment.

Persisters are subpopulations of bacterial communities that stochastically enter a phenotypically dormant state and, therefore, can tolerate antibiotic killing (33). The formation of persisters of *S. aureus* was reported to be associated with a reduction in the level of intracellular ATP (34). We asked whether CDCAA could be applied to this particular population during chronic infections. As expected, CDCAA was able to efficiently remove persisters from *S. aureus* (Fig. 4F).

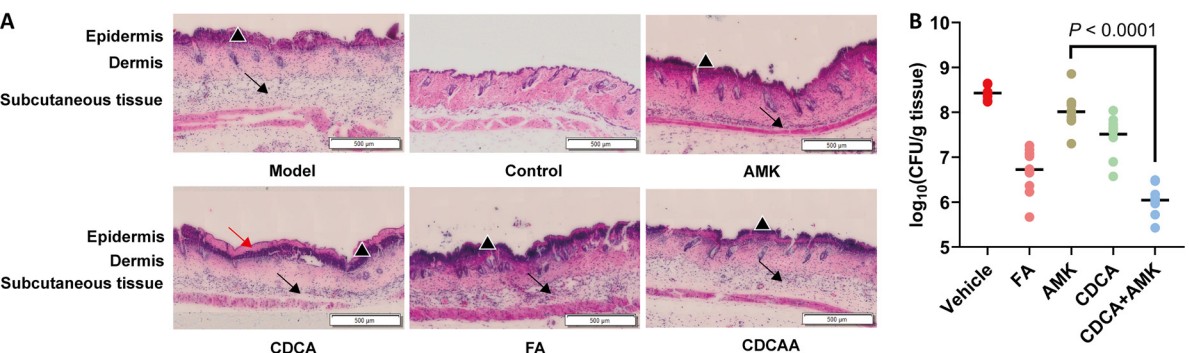

**FIG 5** CDCAA is effective in a mouse skin infection model. (A) Representative histological images of mouse skins. The black triangles indicate neutrophil-infiltrated bands, the black arrows indicate that the subcutaneous tissue is infiltrated with lymphocytes and neutrophils, and the red arrows indicate dermal collagen fibers. The scale bar is 500 $\mu$m. (B) The number of CFU per gram of tissue was determined. Significance was tested by one-way ANOVA in the ranks with the Kruskal-Wallis test ($P < 0.0001$, AMK versus the CDCA+AMK group).

Collectively, we have established that CDCAA could target clinically relevant *S. aureus* populations, including MRSA biofilms, SCVs, and persisters.

**CDCAA rescues a mouse model from *S. aureus* infection.** The efficacy of CDCAA was evaluated using a mouse model *in vivo* using multiple parameters, including histopathology and CFU density *in situ*. As can be seen in Fig. 5A, we observed that compared to the model group, the inflammation induced by MRSA infection was relatively alleviated in the CDCAA group, although the infiltrating bands were still visible in all treatments. Furthermore, CDCAA treatment significantly reduced the bacterial load (2.4 logs) within the *S. aureus* infected skin section compared to individual treatments such as AMK or CDCA (Fig. 5B). As a control, FA also demonstrated a relatively potent effect against *S. aureus* infections (1.7 logs). Consistent with these results, the expression of all serum immune factors, including interleukin-1$\alpha$ (IL-1$\alpha$), IL-1$\beta$, IL-6, IL-8, and KC (keratinocyte-derived chemokine), was reduced compared with the vehicle control (Fig. S4).

**Synergistic mechanisms of the CDCA-aminoglycoside combination.** Since we have established a CDCAA strategy to potentiate aminoglycoside antibiotic killing against *S. aureus*, we next sought to uncover the synergistic mechanisms involved in this process. First, we observed that CDCA at 40 $\mu$g/mL or amikacin alone and the CDCAA combination did not influence the growth of cell line HEI-OC1 (Fig. S5), suggesting the safety of CDCAA. Subsequently, we measured intracellular accumulation of aminoglycoside using fluorescence-labeled aminoglycosides after CDCA treatment and found that internalization of aminoglycosides in *S. aureus* was enhanced in a dose-dependent manner of CDCA (Fig. 6A and Fig. S6). To explain this, we further evaluated membrane permeability using 1-*N*-phenylnapthylamine (NPN) as an indicator and found that CDCA or UDCA increased the membrane permeability of *S. aureus* cells in a dose-dependent manner (Fig. 6B).

Interestingly, we noticed the increase of membrane permeability caused by CDCA is not comparable to that from the increase in aminoglycoside accumulation. We found that CDCA increased the cell membrane permeability by about 1.16-fold at the concentration of CDCA (40 $\mu$g/mL) compared to the control group. However, the uptake of aminoglycosides increased about 6.68-fold compared with the control. We therefore reasoned other mechanisms might exist. It is reported that proton motive force (PMF) is required for aminoglycoside uptake in exponentially growing bacteria (35). Besides, PMF consists of a chemical component ($\Delta$pH) and electrical component ($\Delta\psi$), which are interdependent (10, 36). We first examined the effect of amikacin or CDCA on $\Delta$pH by shifting the external pH from an acidic to an alkaline environment (pH 5.0 to 9.0). We found that the killing activity of amikacin was more effective under alkaline conditions, where $\Delta\psi$ is the main component of PMF (128-fold) (Fig. 6C). Meanwhile, CDCA demonstrated the growth-inhibitory effect on *S. aureus* when the external pH is shifted to acidic conditions, where $\Delta$pH becomes the dominant component of PMF (128-fold) (Fig. 6D). To evaluate the synergy between CDCA and amikacin under different pH

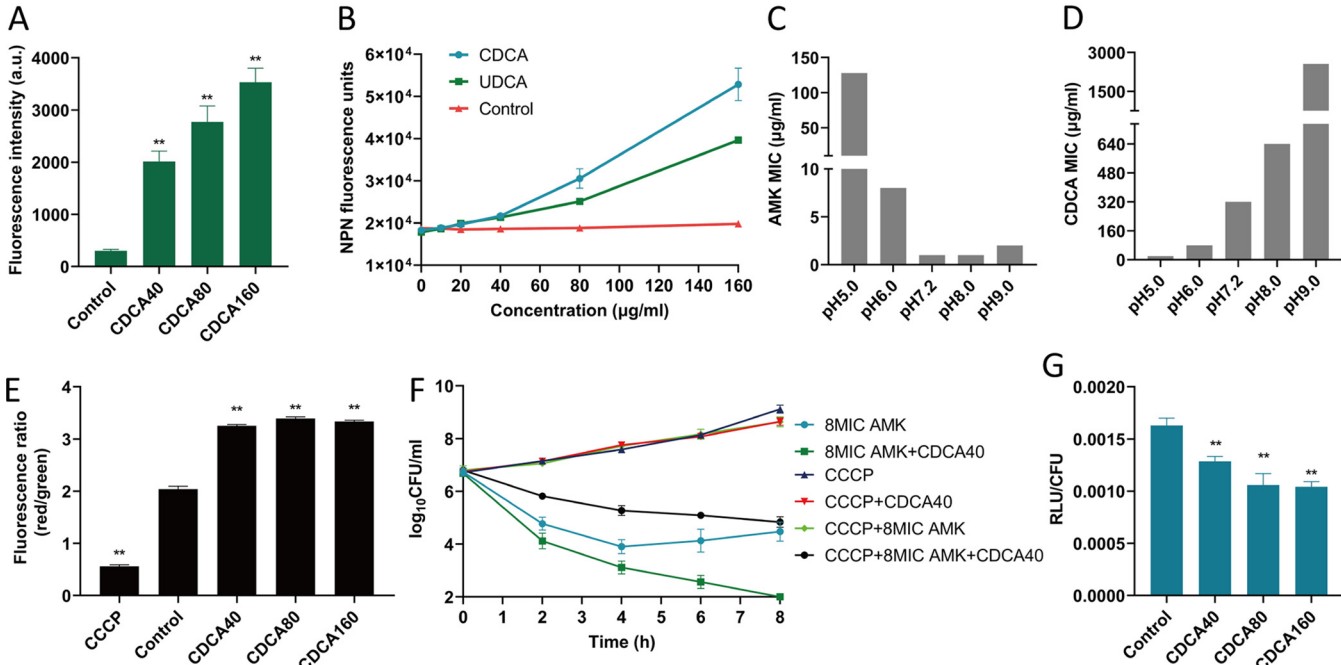

**FIG 6** Synergistic mechanisms of the CDCA-aminoglycoside combination. (A) Texas Red-gentamicin was added to MRSA strain ATCC 43300 cultures with or without different concentrations of CDCA. After 1 h, Texas Red-gentamicin uptake was measured. The error bar represents the mean $\pm$ SEM. Significance was tested by one-way ANOVA (**, $P < 0.0001$). (B) Membrane permeability was assessed by measuring the fluorescence of 1-*N*-phenylnaphthylamine (NPN) after 1 h of exposure to increasing concentrations of CDCA and UDCA. The error bar represents the mean $\pm$ SEM. (C) Effect of altered external pH on the MIC of amikacin. The growth-inhibitory activity of amikacin is enhanced when the external pH is shifted from an acidic to alkaline setting, where $\Delta\psi$ becomes the main component of PMF. (D) Effect of altered external pH on the MIC of CDCA. The growth-inhibitory activity of CDCA is elevated when the external pH is shifted from alkaline to acidic setting, where $\Delta$pH becomes the main component of PMF. (E) The membrane potential of MSSA strain ATCC 25923 was measured by calculating the red/green ratio using the mean fluorescence intensities (MFI) of the population with 5 mM CCCP as a control. The error bar represents the mean $\pm$ SEM. Significance was tested by one-way ANOVA (**, $P < 0.0001$). (F) Comparison of bactericidal ability after 1 $\mu$M CCCP treatment with exponential-phase populations of MSSA strain ATCC 25923. Strains were challenged with 8 $\mu$g/mL amikacin with or without 40 $\mu$g/mL CDCA. At the indicated time points, an aliquot was removed, washed, and plated to enumerate survivors. All experiments were performed in biological triplicate. Error bars represent mean $\pm$ SEM. (G) ATP levels of MRSA ATCC 43300 were measured after treatment with amikacin together with or without different concentrations of CDCA for 6 h. Significance was tested by one-way ANOVA (**, $P < 0.05$).

conditions (pH 5.0 to ~9.0), we performed a checkerboard analysis and revealed that the lower the pH tested (5.0), the more potent the effect of CDCA would be (Fig. S7). Meanwhile, the higher the tested pH (9.0), the more effective amikacin would be, in accordance with the previous conclusion (37).

To further examine whether CDCA had an effect on the $\Delta\psi$ of PMF, we monitored cytoplasmic membrane depolarization (Fig. 6E). We found that CDCA (40, 80, and 160 $\mu$g/mL) used in CDCAA treatment augmented $\Delta\psi$ under normal conditions, supporting our speculation that CDCA did not disrupt the membrane but dissipated $\Delta$pH, causing a compensatory increase of $\Delta\psi$, and finally promoted the influx of amikacin. It was reported that carbonyl cyanide 3-chlorophenylhydrazone (CCCP) is a protonophore and uncoupling agent that causes collapse of PMF and prevents aminoglycoside uptake (38). Treatment with CCCP resulted in abolishment of killing by amikacin, confirming the requirement of PMF for aminoglycoside killing (Fig. 6F). In contrast, the CDCAA group antagonized the action of CCCP by augmenting $\Delta\psi$, thus restoring a certain uptake of amikacin, which implies that it facilitates $\Delta\psi$-dependent aminoglycoside uptake and sensitizes PMF-depleted populations to the killing of aminoglycosides (Fig. 6F). Since PMF is the driving force of ATP synthesis (39), intracellular ATP levels were significantly reduced in the CDCA treatment group compared to the control group (Fig. 6G).

Collectively, our results demonstrated that CDCA promoted intracellular accumulation of aminoglycosides by dissipating $\Delta$pH.

**CDCA potentiates ROS production in *S. aureus*.** To better understand the molecular mechanisms underlying the CDCA-mediated synergy against MRSA, we next analyzed the transcriptome of CDCA-treated *S. aureus* cells. We hypothesized that CDCA

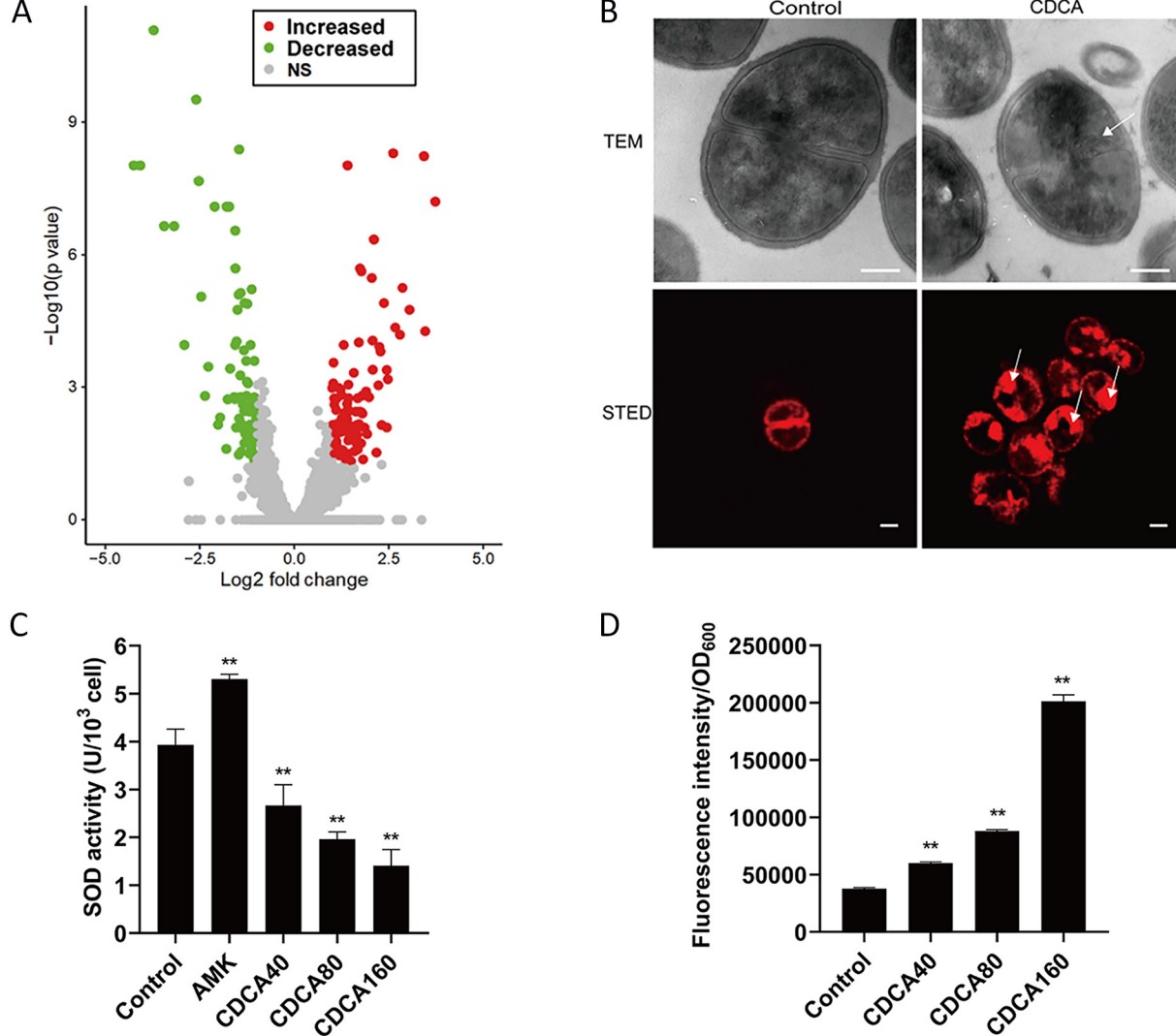

**FIG 7** Transcriptome analysis revealed enhanced ROS production induced by the combination of CDCA and aminoglycosides. (A) Volcano plot of differentially expressed genes in CDCA-treated *S. aureus* cells. NS, not significant. (B) Microscopy analysis of CDCA-treated *S. aureus* cells. Cells were labeled with Nile Red (red fluorescence) (bottom panel). The arrows indicate abnormal membrane topology. The scale bar is 200 nm. (C) SOD levels were measured by calculating the absorbance at 560 nm. The error bar represents the mean ± SEM. Significance was tested by one-way ANOVA (**, $P < 0.05$). (D) ROS levels measured by measuring the fluorescence intensity. The error bar represents the mean ± SEM. Significance was tested by one-way ANOVA (**, $P < 0.01$).

might have an impact on the MRSA physiology that would further potentiate the killing effect of aminoglycosides. Therefore, we performed a transcriptomic analysis of exponential-phase *S. aureus* cells treated with CDCA. RNA was extracted and processed according to the Illumina system for transcriptome sequencing (RNA-seq) analysis. After data processing, a comprehensive data set containing differential gene expression was obtained (Table S3). Treatment with CDCA influenced the expression of 146 genes: that is, 85 genes were upregulated ($P < 0.01$) and 61 downregulated ($P < 0.01$) (Fig. 7A). Functional classification of these genes with altered expression revealed that genes involved in cell structure, metabolism, signal transduction, and genetic information processing showed significant changes at the transcriptional level in CDCA-treated cells.

Specifically, as can be seen from Table S3, we found that genes involved in transport, signal transduction, and virulence were downregulated. First, the expression of genes involved in antibiotic resistance, such as the two-component system PhoP/PhoR, MarR type regulator, and ABC-type transporters, was repressed upon CDCA treatment, suggesting a

susceptible phenotype would occur. Second, expression of *sdrC*, a molecular determinant of *S. aureus* biofilms (40), was inhibited by CDCA treatment, partially explaining the synergy against MRSA biofilms. To further observe the direct effect of CDCA on the membrane, we used stimulated emission depletion microscopy (STED) to examine the integrity of the membrane and found that CDCA could lead to a less uniformly distributed cell content than that in the untreated control (Fig. 7B).

The most remarkable finding is the suppression of *sodA1*, which encodes superoxide dismutase (SOD) in *S. aureus*, leading us to hypothesize that CDCA could alter the antioxidant response in *S. aureus* and that CDCA could act as an ROS potentiating agent to defend against infections with *S. aureus* (Table S3). Furthermore, we found that *ahpF*, *ahpC*, and *dps* were all upregulated upon CDCA treatment, suggesting that these genes were functional upon CDCA addition and may overcome the ROS potentiation effect of CDCA. Based on these facts, we aimed to measure the SOD activity in CDCA-treated cells and found that CDCA significantly decreased SOD activity in a dose-dependent manner (Fig. 7C). On the contrary, when we examined the relative level of reactive oxygen species (ROS) in cells treated with CDCA, the ROS level was found to increase in the presence of CDCA (Fig. 7D), thus explaining the malfunction of the antioxidant system in *S. aureus*.

**CDCA specifically sensitizes Gram-positive pathogens to aminoglycoside.** Having shown that the CDCAA strategy can enable aminoglycoside killing against *S. aureus*, we sought to determine whether a similar phenomenon could be observed in other G$^+$ bacteria. As can be seen in Fig. 8A, we found that CDCAA did not show a synergistic effect against G$^-$ bacteria, including *Acinetobacter baumannii*, *Escherichia coli*, *Klebsiella pneumoniae*, and *Pseudomonas aeruginosa*, indicating that it could function as a G$^+$-specific strategy. We further investigated several G$^+$ pathogens and observed that CDCA could synergize with amikacin for *Enterococcus faecalis*, *E. faecium*, *Staphylococcus epidermidis*, and *Streptococcus pneumoniae*, consistent with our findings for *S. aureus* (Fig. 8B), and the FICI for each G$^+$ pathogen was calculated to be less than 0.5 (Fig. 8C). Thus, CDCA specifically sensitizes G$^+$ pathogens to aminoglycoside killing.

## DISCUSSION

Antibiotic resistance has received tremendous attention due to the paucity of development of novel antibiotics (41). Although chemical biologists have been working hard to find out more alternative antibiotic synthesis pathways, the urgent need for new antimicrobial compounds is now drawing the attention of the WHO (42, 43). They have published lists of priority pathogens for which new antimicrobial agents are in immediate need. To tackle this problem, several scientists are looking for solutions that could slow down the emergence of antibiotic resistance and simultaneously repurpose the approved drugs from the current drug list to extend their inhibitory spectrum (11, 44, 45). In our lab, we attempt to utilize the repertoire of traditional Chinese medicine (TCM) to screen for alternative agents that could be used in combination with antibiotics to control the crisis of multiple-drug resistance (14, 30, 46–48).

Herein, CDCA was shown to be one of the effective components of the TCM formula Tanreqing (TRQ), which could be used in combination with aminoglycosides. This combined strategy, termed CDCAA, demonstrated synergistic potency against G$^+$ bacteria, especially *S. aureus*, *in vitro* and *in vivo*. Furthermore, we elucidated its mode of action against antibiotic-resistant infections by potentiating uptake in a PMF-dependent pathway by dissipating ΔpH, thus sensitizing bacteria to antimicrobial killing with elevated intracellular ROS production. In addition, *in vivo* murine model analysis of this strategy clearly showed that CDCAA protected animals from bacterial infections. Interestingly, this synergy seems to be specific for G$^+$ pathogens and provides an effective treatment against infections caused by G$^+$ bacteria.

**CDCAA as a common strategy against G$^+$ pathogens.** Several promising antimicrobial agents have been reported to target *S. aureus*. With respect to MRSA, several compounds, such as berberine, tiliroside, and silybin, have been isolated as treatment against bacterial infections. However, no studies concerning CDCA had been reported.

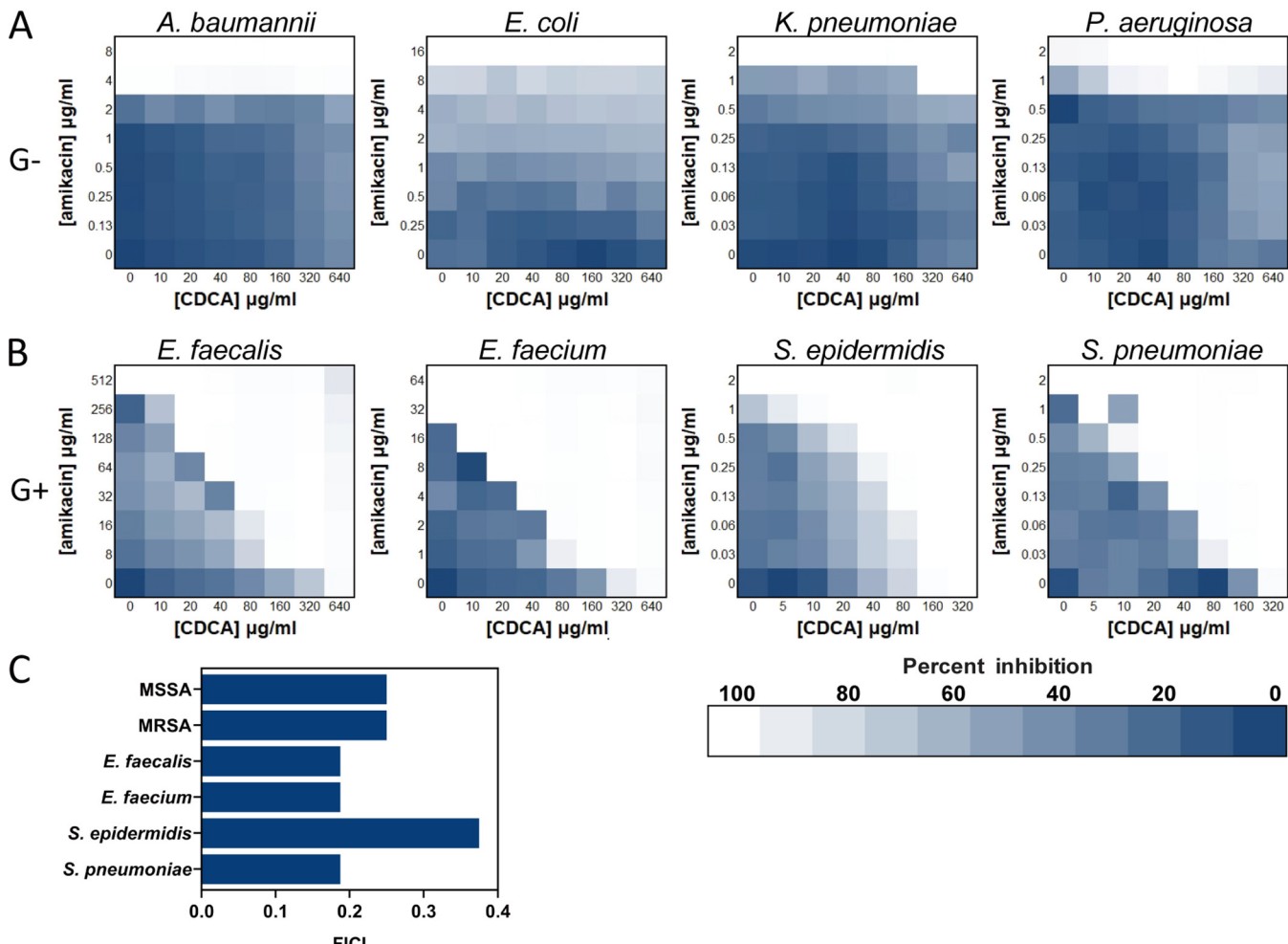

**FIG 8** Common strategy against Gram-positive pathogens by CDCAA. (A) Synergism between CDCA and aminoglycosides evaluated against four G⁻ bacteria—*Acinetobacter baumannii*, *Escherichia coli*, *Klebsiella pneumoniae*, *Pseudomonas aeruginosa*; (B) synergism between CDCA and aminoglycosides evaluated against four G⁺ bacteria—*Enterococcus faecium*, *Enterococcus faecium*, *Staphylococcus epidermidis*, and *Streptococcus pneumoniae*; (C) FICI of amikacin in combination with CDCA calculated from panel B.

Therefore, we discovered a novel effective compound in TCM that reverses aminoglycoside resistance in *S. aureus*. Derivatives of CDCA have also been observed to have similar effects on treatment of resistant pathogens. For example, we have clearly demonstrated UDCA is a relatively less potent analog than CDCA, although UDCA represents a higher content in TRQ than CDCA (49). It is conceivable that both CAs would contribute to synergy against *S. aureus* infections when using TRQ in the presence of aminoglycosides. Furthermore, we have noticed that another CA derivative, isoalloLCA, demonstrated a prominent antimicrobial ability against G⁺ pathogens such as *Clostridioides difficile* and *Enterococcus faecium*, suggesting that these specific bile acid metabolites could reduce the risk of pathobiont infection (22), further strengthening our conclusion that CAs are effective agents in defense against bacterial insults.

Previously, we reported that TRQ could effectively eradicate G⁻ pathogens such as *P. aeruginosa* by disruption of quorum sensing systems and found that Huang Qin (*Scutellariae radix*) played an important role in this process (14). In this study, we focus on Xiong Dan (Ursi Fel), the other main component in TRQ and screened out CDCA as an effective synergistic agent when used in combination with aminoglycoside antibiotics. We elucidated the underlying synergistic mechanisms with aminoglycosides to eradicate G⁺ pathogens, such as *S. aureus* and other species, including *Enterococcus faecalis*, *E. faecium*, *S. epidermidis*, and *Streptococcus pneumoniae*, leading to the conclusion that the CDCAA strategy specifically targets G⁺ pathogens. Collectively, TRQ

has shown its great potential in treating both $G^-$ and $G^+$ pathogens via interfering with distinct pathways in bacteria. The use of TRQ in clinical settings also provides an alternative treatment strategy for mixed infections (50).

**CDCAA as an efficient way to control bacterial infections.** Reversion of antibiotic resistance has been extensively studied (47). For example, rhamnolipids (RL) of *Pseudomonas aeruginosa* were shown to exert a synergistic effect against *S. aureus* for the killing of PMF-independent aminoglycosides (38). We have noticed that RL/aminoglycoside therapy targeted both tolerant and resistant bacterial populations, with the exception of *Streptococcus pneumoniae*, a $G^+$ pathogen, suggesting that RL may function in a different mechanism from that of CDCA. In our study, we indeed observed a distinct mode of action of CDCAA in that CDCA acted as a PMF-dependent synergistic agent by dissipating the chemical potential ($\Delta$pH). In addition, several other effective agents from natural products, such as manuka honey, have shown synergistic effect with oxacillin to sensitize MRSA to oxacillin by downregulating *mecR1*, the regulator of *mecA*, which encodes the low-affinity penicillin binding protein 2a (PBP2a) (51, 52).

In addition to reversion of antibiotic resistance in $G^+$ pathogens, countermeasures have been extensively studied to expand their applications in infection prevention, such as in $G^-$ pathogens (37, 53) and *Mycobacterium* (54). The reason to repurpose these approved drugs or perform the screening in natural products stems from our shortage of novel antibiotics and the rapid development of antibiotic resistance. Therefore, it is extremely important to search for more alternative combinatory approaches to combat these infections caused by $G^+$, $G^-$, and other types of pathogens (10). This combinatory therapy will greatly reduce the amount of antibiotics used in clinics and alleviate the rise of antibiotic resistance (55). Our study has provided fundamental insights into the potential use of effective components in TCM when using antibiotics.

**CDCAA-induced antioxidative response as a promising target for antibacterial killing.** There is a common mechanism used by three major classes of bactericidal antibiotics that induce the production of highly deleterious hydroxyl radicals in both $G^+$ and $G^-$ bacteria, which eventually lead to cell death (56). Further study illustrated that enhanced production of endogenous microbial ROS could potentiate antibacterial activity of antibiotics (57). Furthermore, targeting bacterial stress response systems, such as stringent response and membrane stress response, could also enhance antibiotic activity (58, 59). Therefore, it is interesting to harness oxidative stress-related pathways as a target to improve bacterial killing by antibiotics (60).

In this study, we have found that CDCA not only could target SOD activity as a measure to enhance endogenous ROS production due to an inability in detoxification, but also could increase the uptake of aminoglycosides and ultimately contribute to the bacterial killing by aminoglycosides. This dual mechanism of CDCAA in the eradication of $G^+$ bacterial infections has provided an ideal example of combinatory antibiotic and nonantibiotic therapy (10, 61). Therefore, the search for ROS-potentiating agents has become an active field in the control of bacterial infection. For example, the use of polymers such as guanidinium-functionalized polycarbonate and poly(2-oxazoline) could significantly enhance the intracellular ROS production and finally lead to cell death (53). The identification of CDCA as an ROS potentiation agent sets an excellent example for the application of TCM in bacterial infections.

**Limitations and conclusions.** There are several limitations in this study. First, we have not been able to directly label amikacin but could use gentamicin to observe the behavior of aminoglycoside uptake in *S. aureus*. Future work will focus on the synthesis of series of fluorescent aminoglycosides to study their cellular distribution by microscopy analysis. Second, we have shown that SOD activity was influenced by CDCA treatment. It would be intriguing to investigate the interaction of CDCA and SOD *in vitro* by further biochemical analysis. Third, the immunological response of CDCAA treatment was not sufficient for the protection of *S. aureus* infections.

In conclusion, we demonstrated that CDCA, an effective component of TRQ, selectively sensitizes *S. aureus* bacteria, including those associated with biofilms, SCVs, and persisters, as well as other $G^+$ pathogens to aminoglycoside killing. CDCA is safe for

mammalian cells, represses resistance development, and is ideal for the reduction of severe use of antibiotics in clinical settings. In the future, we would like to perform clinical trials of CDCA as a treatment against bacterial infections and expand its antibacterial spectrum of clinical significance.

## MATERIALS AND METHODS

**Strains and culture conditions.** The bacterial strains used in this study are listed in Table S4 in the supplemental material. Strains were grown under standard culture conditions at 37°C in either Mueller-Hinton broth (MHB) or lysogeny broth (LB). Bacterial isolates were collected from the clinical laboratory of Dongzhimen Hospital and Guanganmen Hospital, Beijing, China.

**Growth curve.** Overnight cultures of bacterial strains in LB were diluted 100-fold in 3 mL fresh LB medium and precultures were incubated aerobically at 37°C in a shaker at 200 rpm to obtain an optical density at 600 nm ($OD_{600}$) of 0.5. The precultures were further diluted 100 times in 96-well microtiter plates containing $10^5$ cells. Wells with LB medium were set as a control. CDCA was prepared in LB medium at concentrations of 40, 80, 160, and 320 $\mu$g/mL. Microtiter plates were incubated for 24 h at 37°C in a Synergy H1 hybrid multimode reader (BioTek) using the following settings: shaking for 30 s every min and absorbance measured every 30 min at 600 nm. Each treatment was performed in triplicate.

**MIC assays.** MICs were determined using broth dilution assays as described previously (30). Briefly, $5 \times 10^5$ CFU/mL were incubated with various concentrations of antibiotics in a total volume of 200 $\mu$L of MHB in a 96-well plate. MICs were determined after incubation at 37°C for 18 to ~24 h.

**Checkerboard synergy assays.** A checkerboard broth dilution assay was performed to calculate the fractional inhibition concentration index (FICI). A total of $5 \times 10^5$ CFU/mL were incubated in a 96-well plate that was distributed to an 8-by-8 matrix by serial 2-fold serial dilutions of each compound added to 2-fold aminoglycoside dilutions. The FICI was calculated as follows: FICI = (MIC of compound A in combination/MIC compound A alone) + (MIC compound B in combination/MIC compound B alone). An FICI value of ≤0.5 was interpreted as synergy, 0.5 < FICI ≤ 1 indicates an additive effect, 1 < FICI ≤ 2 indicates no interaction, and 2 < FICI indicates antagonism.

**Membrane potential measurements.** Bacterial membrane potential was tested using the BacLight bacterial membrane potential kit as per the manufacturers' instructions. Overnight bacterial cultures were diluted 100-fold in 3 mL MH medium and then grown to exponential phase ($OD_{600} \approx 0.6$) before a 30-min exposure to the indicated concentration of CDCAs. Cultures were then diluted in 1 mL of HEPES and treated with 30 mM $DiOC_2$ for 15 min. Analysis on a flow cytometer (Becton, Dickinson) was recorded, and relative membrane potential was calculated by taking the ratio of population red and green linear mean fluorescence intensity (MFI) values.

**Membrane permeabilization assay.** Cell permeabilization assays were performed using 1-*N*-phenylnaphthylamine (NPN) fluorescence as an indicator of outer membrane permeability as previously described with modifications (62). A Synergy H1 hybrid multimode reader (BioTek) was used with filters of 355 nm for excitation and 405 nm for emission. Assays were performed in black with clear-bottom 96-well plates and read at 30 min at room temperature. Ninety-six-well plates were supplemented with 100 $\mu$L of bacterial suspension ($10^6$ CFU) in 5 mmol/L HEPES buffer (pH 7.2), 50 $\mu$L of 40 $\mu$mol/L NPN, and CDCA in 50 $\mu$L of buffer. The results were expressed as relative fluorescent units, representing the fluorescence value of the bacterial suspension with the CDCAs and NPN subtracted from the corresponding value of the bacterial suspension and NPN without the CDCAs.

**Time-kill curve assay.** Time-kill analysis was used to determine *S. aureus* growth in the presence of various antibiotics, used as monotherapies or in combination. Where indicated, a bacterial suspension was added to each solution ($5 \times 10^6$ CFU/mL for 5 mL), and incubated at 37°C with shaking. Growth control flasks contained only bacteria and 5 mL of MH broth. Each aliquot was serially diluted, seeded on MH agar plates and incubated for 24 h at 37°C in order to determine the number of CFU per milliliter. The colonies were counted only on plates that had 30 to 300 colonies. Subsequently, time-kill curves were generated by plotting $log_{10}$ CFU per milliliter against time (hours).

**Biofilm susceptibility assays.** The preparation of a mature MRSA biofilm was carried out according to our previous work (15). Briefly, overnight MRSA culture in LB medium supplemented with 0.25% glucose (LB-G) was diluted in the respective medium to an $OD_{600}$ of 0.05. Subsequently, 200-$\mu$L aliquots of bacterial solutions were aliquoted into a 96-well microtiter plate. Wells containing medium served as negative controls. After 24 h of incubation at 37°C, medium and planktonic cells were removed and wells containing biofilms were rinsed with 200 $\mu$L of 0.9% NaCl. Antibiotic agents diluted in LB-G medium at different concentrations were tested as single drugs and in combination by adding 200 $\mu$L of the tested sample to each well. Wells containing medium only served as negative controls. After 24 h of incubation at 37°C, the wells were washed to remove planktonic cells and ensure that only cells within biofilms were tested. Then the XTT [2,3-bis-(2-methoxy-4-nitro-5-sulfophenyl)-2H-tetrazolium-5-carboxanilide salt] reduction assay was performed to evaluate the viability of the biofilms. For this purpose, 40 $\mu$L of XTT-phenazine methosulfate (PMS) solution (200 mg/mL XTT, 2 $\mu$M PMS) was added to each well, plates were incubated for 2 h at 37°C in the dark, and the optical density was then measured at 490 nm using a Synergy H1 hybrid multimode reader (BioTek). A decrease in the number of live cells correlates with the reduction in the overall activity of the dehydrogenases responsible for the transformation of the sodium salt of tetrazolium XTT into formazan, which was determined colorimetrically. All experiments were repeated three times. The biofilm structures were visualized using a confocal laser scanning microscope (Olympus FV1000) and processed with Imaris (v.9.3.1; Bitplane). The bacteria were

then treated with or without CDCA in a confocal plate at a concentration of 40 $\mu$g/mL for 24 h in an incubator at 37°C, stained using a LIVE/DEAD BacLight bacterial viability kit, and incubated for another 24 h. SYTO 9 was excited at 488 nm, whereas propidium iodide was excited at 561 nm.

**Evolution of resistance.** Resistance evolution was tested for drug-treated *S. aureus* and MRSA after 30 days of passaging in MH medium. Cells were passed every 24 h. To determine the MICs of antibiotics alone and drug combinations, the wells with the highest concentration of antibiotics that permitted significant bacterial growth ($OD_{600} \geqq 0.1$) were used to inoculate fresh MHB for the next passage at a bacterial density of ~5 × $10^5$ CFU/mL. At every third passage, strains were collected and stored at −80°C in a bacterial cryopreservation tube.

**SCV and persisters of *S. aureus* strains.** SCVs were generated during the bacterial culturing experiment and collected for matrix-assisted laser desorption ionization–time of flight mass spectrometry (MALDI-TOF MS) identification. We further verified isolated SCVs using selective medium supplemented with hemin, menadione, and thymidine to determine auxotrophy of SCVs.

Persister cells of MRSA ATCC 43300 were grown overnight in LB medium and then diluted in fresh medium and incubated until the cells reached the exponential phase of growth. Bacteria were then exposed to 10× MIC of ciprofloxacin (CIP) for 6 h, followed by spinning down and washing the bacteria pellet with sterile 1× phosphate-buffered saline (PBS) twice. Then, the bacterial pellet was resuspended in fresh PBS and different treatments were applied.

**Gentamicin-Texas Red intracellular uptake assay.** Gentamicin was dissolved in 0.1 M $NaHCO_3$ buffer (pH 8.5) at a final concentration of 10 mg/mL. Texas Red sulfonyl chloride (T1905; Thermo Fisher Scientific) was dissolved in *N,N*-dimethylformamide at a final concentration of 20 mg/mL. On ice, 50 $\mu$L Texas Red was added slowly to 2 mL gentamicin solution. The mixed solution was then stirred at room temperature for 1 h. The ATCC 43300 strain was grown to the mid-exponential phase and then treated with different concentrations of CDCA and gentamicin-Texas Red at a final concentration of 32 $\mu$g/mL for 1 h. After treatment, bacteria were collected by being washed with sterile 1× PBS twice. A Synergy H1 hybrid multimode reader (BioTek) was used with filters of 589 nm for excitation and 615 nm for emission. Assays were performed in black clear-bottom 96-well plates and read at room temperature. Visualization of gentamicin-Texas Red on bacterial cells was achieved using a confocal laser scanning microscope (Olympus FV1000) with an excitation wavelength of 589 nm, and the image was processed with Imaris (v.9.3.1; Bitplane).

**ATP assays.** *S. aureus* ATCC 43300 was grown to ~3 × $10^8$ CFU/mL in 3 mL of MHB and treated with CDCAA for 6 h (40 $\mu$g/mL CDCA). ATP levels were measured using a Promega BacTiter Glo kit according to the manufacturer's instructions.

**SOD activity.** *S. aureus* ATCC 43300 overnight cultures were grown to 5 × $10^8$ CFU/mL in 10 mL of MHB containing AMK or CDCA. Cultures were incubated at 37°C with rotation (225 rpm). Superoxide dismutase (SOD) levels were measured using a SOD assay kit (BC0175; Solarbio) according to the manufacturer's instructions. The absorbance was measured at 560 nm using a Synergy H1 hybrid multimode reader (BioTek).

**ROS.** *S. aureus* ATCC 43300 overnight cultures were grown to 5 × $10^8$ CFU/mL in 10 mL of MHB with or without CDCA. The bacteria were collected by being washed with sterile 1× PBS twice and then treated with 32 $\mu$g/mL AMK for 4 h. After treatment, bacteria were collected by being washed with sterile 1× PBS twice. ROS levels were measured using an ROS assay kit (CA1410; Solarbio) according to the manufacturer's instructions. The fluorescence intensity was measured with excitation at 488 nm and emission at 525 nm using a Synergy H1 hybrid multimode reader (BioTek).

**Labeling and imaging of *S. aureus* membrane.** To visualize the membrane in *S. aureus* by stimulated emission depletion microscopy (STEDYCON; Abberior), cells from cultures at 5 × $10^7$ CFU/mL were stained with Nile Red at a final concentration of 10 $\mu$g/mL. The cells were incubated at 30°C with agitation for 10 min. Unbound dye was removed from the medium by washing cells with 1× PBS and placed 5 $\mu$L on 50 $\mu$L Abberior Mount Solid antifade. Cells showing uniform Nile Red labeling were imaged using a 45% 561-nm laser for excitation, 100% 775-nm laser for STED, and emission at 616 nm.

**Cell culture.** The HEI-OC1 (House Ear Institute-organ of corti 1) cells were cultured in Dulbecco's modified Eagle's medium (DMEM) containing 10% fetal bovine serum (FBS) (Gibco, Australia) with 5% $CO_2$ at 37°C in a humidified incubator.

**Cytotoxicity assays.** Cytotoxicity activity was measured for HEI-OC1 cells using the CCK-8 (Cell Counting Kit-8) system (Dojindo, Japan) according to the manufacturer's instructions. Cells were seeded into 96-well plates at a density of 8,000 cells per well. At 24 h after seeding, after treatment with various concentrations of antimicrobials, cell viability was measured after another 24 h, 10 $\mu$L CCK-8 solution was added to each well, and the plate was incubated at 37°C for 1 h in the dark. The absorbance was measured at 450 nm with a Synergy H1 hybrid multimode reader (BioTek).

**Transcriptome sequencing.** RNA degradation and contamination were monitored on 1% agarose gels. RNA integrity was evaluated using the RNA Nano 6000 assay kit of the Bioanalyzer 2100 system (Agilent Technologies, CA, USA). Total RNA was used as input material for the preparation of RNA samples. For prokaryotic samples, mRNA was purified from total RNA using probes to remove rRNA. Fragmentation was performed using divalent cations at an elevated temperature in the first-strand synthesis reaction buffer (5×). First-strand cDNA was synthesized using a random hexamer primer and Moloney murine leukemia virus (MMLV) reverse transcriptase, and then RNase H was used to degrade the RNA. In addition, in the DNA polymerase I system, dUTP was used to replace the deoxynucleoside triphosphate (dNTP) of dTTP as the raw material to synthesize the second strand of cDNA. The remaining overhangs were converted to blunt ends via exonuclease/polymerase activities. After adenylation of the 3′ ends of the DNA fragments, the adaptor with the hairpin loop structure was ligated to prepare for

hybridization. Then the USER enzyme was used to degrade the second strand of cDNA containing U. To preferentially select preferentially 370- to ~420-bp cDNA fragments, library fragments were purified with the AMPure XP system (Beckman Coulter, Beverly, MA, USA). Then, PCR was performed with Phusion high-fidelity DNA polymerase, universal PCR primers, and index (X) primer. Finally, the PCR products were purified with the AMPure XP system, and the library quality was evaluated on the Agilent Bioanalyzer 2100 system. Clustering of the index-coded samples was performed on a cBot cluster generation system using the TruSeq PE (paired-end) cluster kit v.3-cBot-HS (Illumina) according to the manufacturer's instructions. After cluster generation, library preparations were sequenced on an Illumina Novaseq platform, and 150-bp paired-end reads were generated.

**Transcriptome data analysis.** Raw data (raw reads) of fastq format were first processed through in-house perl scripts. In this step, clean data were obtained by removing reads containing adapter, reads containing N base, and low-quality reads from raw data. At the same time, for Q20 and Q30 quality scores and GC content, the clean data were calculated. All downstream analyses were based clean and high-quality data. The read mapping to the reference genome and the gene model annotation files were downloaded directly from the genome website. Both the building index of reference genome and alignment of clean reads to the reference genome were performed with Bowtie 2-2.2.3. Differential expression analysis of two conditions/groups (three biological replicates per condition) was performed using the DESeq R package (1.18.0). DESeq provides statistical routines to determine differential expression in digital gene expression data using a model based on the negative binomial distribution. The resulting $P$ values were adjusted using the Benjamini and Hochberg's approach to control the false-discovery rate.

**Animal usage declaration.** Eight-week-old male BALB/c mice were obtained from Beijing Vital River Laboratory Animal Technology Co., Ltd. (no. CNAS LA0004). The mice were kept strictly according to the regulations for the administration of experimental animals approved by the State Council of the People's Republic of China (GB/T 35892-2018) and the technical specifications of the Ethical Review for Laboratory Animal Welfare (DB11/T 1734-2020). The animal study protocols were performed in accordance with the relevant guidelines and regulations [SYXK(Beijing)2021-0017, Experimental Research Center, China Academy of Chinese Medical Sciences]. The laboratory animal usage license number is SCXK(Beijing)2016-0006, certified by Beijing Vital River Laboratory Animal Technology Co., Ltd.

**Mouse skin infection model.** Mice were adapted to standardized environmental conditions (temperature, 23 $\pm$ 2°C; humidity, 55% $\pm$ 10%) for 36 h before infection. Briefly, anesthesia was induced with 5% isoflurane and maintained at 2.5%. An area of ~2 cm$^2$ was removed in the dorsal region of the neck was stripped using 25 autoclave tape strips (3M) until the skin was visibly shiny but not bleeding. For the bacterial preparation, *S. aureus* ATCC 43300 was grown overnight in LB medium and subcultured to an $OD_{600}$ of 0.5. The bacteria were adjusted to 5 $\times$ 10$^7$ CFU/mL, and a 20-$\mu$L volume (10$^6$ CFU) was applied to the wounded area. Test antibiotics were formulated in sodium carboxymethyl cellulose (CMC-Na) at 1% (wt/wt), and the animals were treated 8 times over a 38-h period at the following times after infection: 2, 8, 14, 20, 26, 32, and 38 h. Each treatment was a 50-mg formulation by weight. The same dose of fusidic acid (FA) was chosen as a positive control. Both model and control mice received 1.5% CMC-Na vehicle, with the exception that the control group had intact epidermis. The mice were sacrificed 1 h after the last dose. Blood plasma samples were collected by the abdominal aortic method. At the endpoint, an ~2-cm$^2$ area of skin was excised and homogenized in 1 mL 0.9% normal saline for 15 min. The homogenates were serially diluted and seeded in MH agar plates and then incubated for 24 h at 37°C to determine the number of CFU per gram of tissue. Four animals (technical replicates) were used in each treatment group, divided among three experiments carried out on independent occasions (biological replicates). Mice were infected as described above, and skins were harvested and fixed in 10% neutral buffered formalin for 48 h and transferred to 70% ethanol. The fixed skins were embedded with paraffin and sectioned at 6 $\mu$m for hematoxylin and eosin staining. In addition, blood plasma samples were centrifuged at 4,000 rpm for 10 min at 4°C, and the supernatants were stored at $-80$°C for cytokine analysis by enzyme-linked immunosorbent assay (ELISA).

**Statistical analysis.** Student's *t* test was used when one-way analysis of variance (ANOVA) revealed significant differences ($P < 0.001$). All statistical analyses were performed with GraphPad Prism statistical software (GraphPad Software, La Jolla, USA) with the assistance of Excel (Microsoft).

**Data availability.** All data are available in the main text and/or the supplemental material. The GEO accession number of the RNA-seq data is GSE193011.

## SUPPLEMENTAL MATERIAL

Supplemental material is available online only.
**SUPPLEMENTAL FILE 1**, PDF file, 2.2 MB.

## ACKNOWLEDGMENTS

We thank the Kaibao Pharmaceutical Company, Shanghai, China, for support. We are grateful for technical assistance from other members of the Wang lab.

We are grateful for the support from the Fundamental Research Funds for the Central Public Welfare Research Institutes (XTCX2021002), Academician Expert Workstation (HX2017001), and National Key Research and Development Program of China (no. 2019YFC1709305).

Conceptualization, Y.W. and Q.W.; Methodology, Q.W., G.H., D.L., G.L., Z.L., S.C., and P.C.; Investigation, K.C., W.Y., S.M., and Y.S.; Visualization, K.C., W.Y., Q.W., Y.C., and X.J.; Supervision, Y.W., Q.W., and W.Y; Writing – Original Draft: Q.W., K.C., and W.Y.; Writing – Review & Editing: Q.W., K.C., W.Y., S.C., P.C., and Y.W.

Experimental Research Center, China Academy of Chinese Medical Sciences, January 18, 2022. China Patent application 202210055969.3.

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
