## [Reviewer comments · Microbiology Spectrum]

Microbiology Spectrum

Chenodeoxycholic acid and amikacin combination enhance eradication of *Staphylococcus aureus*

Kaiyu Cui, wei yang, Zhiyuan Liu, Guijian Liu, Dongying Li, Yanan Sun, Gaiying He, Shuhua Ma, Yu Cao, Xuefan Jiang, Sylvie Chevalier, Pierre Cornelis, Qing Wei, and Yi Wang

Corresponding Author(s): Qing Wei, Nanchang Institute of Technology

Review Timeline:

Submission Date:	July 5, 2022
Editorial Decision:	September 13, 2022
Revision Received:	October 18, 2022
Editorial Decision:	November 15, 2022
Revision Received:	November 25, 2022
Accepted:	December 7, 2022

Editor: Cristina Solano

Reviewer(s): Disclosure of reviewer identity is with reference to reviewer comments included in decision letter(s). The following individuals involved in review of your submission have agreed to reveal their identity: Renee Marie Fleeman (Reviewer #2)

Transaction Report:

DOI: <https://doi.org/10.1128/spectrum.02430-22>

September 13, 2022

Dr. Qing Wei
Nanchang Institute of Technology
Yingxiong Road
Nanchang, Jiangxi 330044
China

Re: Spectrum02430-22 (Chenodeoxycholic acid and amikacin combination enhance eradication of *Staphylococcus aureus*)

Dear Dr. Qing Wei:

Thank you for submitting your manuscript to Microbiology Spectrum. It has now been reviewed by 2 experts in the field who consider the topic to be interesting, though one of the reviewers expressed substantive criticisms of a number of aspects of the article that require major changes. Their comments are provided at the bottom of this letter. Taking these comments into account, I wish to consider a revised version of the manuscript, substantially improved, that adequately addresses all criticisms and concerns expressed by both reviewers. Also, the manuscript should be edited for English language, since the reviewers indicated problems relating to language correctness and clarity.

Link Not Available

Sincerely,

Cristina Solano

Journals Department
Reviewer comments:

Reviewer #1 (Comments for the Author):

The manuscript by Cui and co-workers have extensively characterized the antimicrobial activity of chenodeoxycholic acid (CDCA, a component of traditional Chinese medicines) against *Staphylococcus aureus*. This compound does not select for resistant mutants in vitro (Figure 3A), but shows a rather low antimicrobial activity (MIC 320 µg/ml; Figure 1A) and at the

concentration required for inhibiting growth of *S. aureus* it results rather toxic for mammalian cells (<20% survival rate; Figure S4). The authors, however, have identified an interesting synergy between CDCA and the aminoglycoside amikacin (Figure 2), with many positive features: the combination of CDCA and amikacin prevents selection of mutants resistant to amikacin (Figure 3B), it is active against variants of *S. aureus* (Figure 4), and it is active in an in vivo model of skin infection (Figure 5), among other characteristics. Unfortunately, amikacin is not among the first line drugs for the treatment of staphylococcal infections, so the clinical relevance of this work is rather low.

The rationale for selecting amikacin as a companion drug to CDCA is not shown (line 133: We focus on amikacin...), so I would suggest to screen whether CDCA also presents synergy with other, more relevant drugs from the clinical point of view. If so, this would increase notably the clinical interest of this work.

In addition, some parts of the manuscript are unclear. For example, the use of certain control drugs or experimental conditions in various experiments should be better explained in order to have a clear idea about the meaning of the outcome of the experiment (for example, in figure 5, what is the difference between model and control?). Also, some sentences are vague, for example, line 252, ...restoring H concentration, where?, in line 175, ...repress the development of detectable mutations, TO AMIKACIN should be added for clarity. It would be good to review all the manuscript for providing enough details for a better comprehension. In lines 229-230, when it says that the increase of membrane permeability caused by CDCA IS NOT COMPARABLE to that of the increase of aminoglycoside accumulation, the rationale behind this assessment should be detailed.

Finally, the size of some figures is too small, thus preventing from proper analysis of data presented.

Reviewer #2 (Comments for the Author):

The authors describe the in-depth characterization of a component of from the traditional Chinese medicine Tanreqing. They have identified this agent synergizes with aminoglycosides to kill *Staphylococcus aureus*. The agent, chenodeoxycholic acid dissipates the proton motor force without disruption of the membrane. The activity against persister development initiated by aminoglycosides is compelling research for future therapeutics. The detailed work describing the physiological effects of chenodeoxycholic acid revealed the mechanism used to synergize with aminoglycosides. With the importance of the findings of this work, minor modifications to the manuscript would greatly benefit readability and the understanding of this important synergistic abilities of chenodeoxycholic acid (Please, see additional comments in the attached file).

Staff Comments:

Preparing Revision Guidelines

Please return the manuscript within 60 days; if you cannot complete the modification within this time period, please contact me. If you do not wish to modify the manuscript and prefer to submit it to another journal, please notify me of your decision immediately so that the manuscript may be formally withdrawn from consideration by Microbiology Spectrum.

The authors describe the in-depth characterization of a component of from the traditional Chinese medicine Tanreqing. They have identified this agent synergizes with aminoglycosides to kill *Staphylococcus aureus*. The agent, chenodeoxycholic acid dissipates the proton motor force without disruption of the membrane. The activity against persister development initiated by aminoglycosides is compelling research for future therapeutics. The detailed work describing the physiological effects of chenodeoxycholic acid revealed the mechanism used to synergize with aminoglycosides. With the importance of the findings of this work, minor modifications to the manuscript would greatly benefit readability and the understanding of this important synergistic abilities of chenodeoxycholic acid.

Major comments:

1. FICI values are commonly reported by adding the FICs of both drugs ($\Sigma\text{FIC} = \text{FIC}_{\text{antibiotic}} + \text{FIC}_{\text{CDCA}}$) and a value of < 0.5 is synergistic. Although the fact that these agents have synergistic activities is clear, only the FIC of the antibiotics are reported. It would be easier if the reviewer did not have to figure out the FIC of CDCA when tested with the antibiotics.
2. The biofilm work in figure 4 shows great promise for CDCAA. However, the effect of amikacin alone or combined with CDCA is not discussed in the text for its ability to disrupt a biofilm or any combinatorial effect seen or not seen with these experiments.

Minor comments:

1. TCM in the abstract is not mentioned what it stands for.
2. There are a lot of acronyms throughout the text. Perhaps remove those that are used less than 5 times in the article to make it easier on the reader. (SCV, EMA, NMPA, FDA, etc.)
3. WHO on line 61 does not say what it stands for
4. Supplemental table 1 has bolded values and it is not clear or stated why they are bolded
5. Line 207 "efficacy or CDCAA efficacy" should be reworded
6. Line 221 Cell line HEI-OCI is not described well enough. This is an uncommon cell line and it should be acknowledged as a mammalian cell line and provided with the reason for this cell line choice over the commonly used kidney and liver cell lines.
7. In the discussion G+ and G- are abbreviated but not anywhere else in the text. It starts in the last section of the results. Please ensure uniformity throughout the text.
8. Line 390 italicize *S. aureus*.
9. The statistical analysis used to find the P values should also be included in the figure legends so the reader can see what equation was used for those values. And there is no mention of corrections used for multiple analyses. The exception to this was Fig 5B that included all information.

Reviewer#1:

The manuscript by Cui and co-workers have extensively characterized the antimicrobial activity of chenodeoxycholic acid (CDCA, a component of traditional Chinese medicines) against *Staphylococcus aureus*. This compound does not select for resistant mutants in vitro (Figure 3A), but shows a rather low antimicrobial activity (MIC 320 µg/ml; Figure 1A) and at the concentration required for inhibiting growth of *S. aureus* it results rather toxic for mammalian cells (<20% survival rate; Figure S4). The authors, however, have identified an interesting synergy between CDCA and the aminoglycoside amikacin (Figure 2), with many positive features: the combination of CDCA and amikacin prevents selection of mutants resistant to amikacin (Figure 3B), it is active against variants of *S. aureus* (Figure 4), and it is active in an in vivo model of skin infection (Figure 5), among other characteristics. Unfortunately, amikacin is not among the first line drugs for the treatment of staphylococcal infections, so the clinical relevance of this work is rather low.

1. The rationale for selecting amikacin as a companion drug to CDCA is not shown (line 133: We focus on amikacin...), so I would suggest to screen whether CDCA also presents synergy with other, more relevant drugs from the clinical point of view. If so, this would increase notably the clinical interest of this work.

Response: Thanks for the critical review and constructive advice. Previously, we have shown that TRQ combined with vancomycin or linezolid against methicillin-resistant *Staphylococcus aureus* in our study (doi: 10.1186/s12906-018-2231-8). In addition, we found that CDCA as an active component of the TRQ had synergistic effects with various clinical antibiotics, including aminoglycosides and β -lactams. The synergistic mechanism of CDCA and aminoglycosides is distinct from other antibiotics, and the combination mechanism of other kinds of antibiotics is being studied successively. By using amikacin as a representative type of aminoglycosides, we aim to highlight the potential of CDCA as an adjuvant in expanding the antibacterial spectrum of AMK and other aminoglycosides.

In addition, we have changed the wording in the text and highlighted them as follows:

“Previously, we have shown that TRQ combined with vancomycin or linezolid exhibited synergy against MRSA (30). In this study, we selected aminoglycoside class antibiotics to investigate their activity against Gram-positive pathogens. As shown in Fig. 2A and 2B, lower dose of CDCA significantly increased the anti-*S. aureus* effect of several aminoglycosides, including amikacin, etimicin, gentamicin, kanamycin and tobramycin in both MSSA and MRSA strains. Based on combinatory efficacy, we focus on amikacin, a potent aminoglycoside against *S. aureus* and determined that MRSA ATCC 43300 showed a MIC value of 64 µg/ml.”

2. In addition, some parts of the manuscript are unclear. For example, the use of certain control drugs or experimental conditions in various experiments should be better explained in order to have a clear idea about the meaning of the outcome of the experiment (for example, in figure 5, what is the difference between model and control?).

Response: Thanks for the critical review and constructive advice. We apologize for the lack of detail and have improved the description by adding more details as follows in the text of Mouse skin infection model part in Materials and Methods: “Both model and control mice received 1.5% CMC vehicle, with the exception that control group was intact in epidermis”.

3. Also, some sentences are vague, for example, line 252, ...restoring H concentration, where?,

Response: We thank the reviewer for pointing out this issue. We have changed the words of “augmenting $\Delta\psi$ ” instead of “restoring H⁺ concentration”. In Figure 6F, we used CCCP as a control to collapse PMF and more precisely, $\Delta\psi$, and to prevent aminoglycoside uptake. However, CDCA could antagonized this activity by augment $\Delta\psi$ (Figure 6E) and CDCAA finally sensitizes PMF-depleted bacterial populations to the killing of aminoglycosides.

4. in line 175, ...repress the development of detectable mutations, TO AMIKACIN should be added for clarity.

Response: Thanks for the comments. We have changed the words “repress the development of detectable mutations” to “repress the development of aminoglycoside resistance”. Since we have noticed that we did not examine the mutations elicited by serial passages of bacteria. However, we indeed measured the MIC of these mutants and thus we used “resistance” instead of “mutations” in the revised text. However, it would be interesting to investigate the mutations underlying these mutants.

5. It would be good to review all the manuscript for providing enough details for a better comprehension. In lines 229-230, when it says that the increase of membrane permeability caused by CDCA IS NOT COMPARABLE to that of the increase of aminoglycoside accumulation, the rationale behind this assessment should be detailed.

Revised Fig. 6B

Response: Thanks for the critical review and constructive advice. We have updated the methods and results of Figure 6B to be consistent with the conditions in Figure 6A. We found that CDCA increased the cell membrane permeability by about 1.16-fold at the concentration of CDCA (40 µg/ml) as compared to the control group. However, the uptake of aminoglycosides increased 6.68-fold compared with control.

In addition, we have modified the results part as follows:

“Interestingly, we noticed the increase of membrane permeability caused by CDCA is not comparable to that of the increase of aminoglycoside accumulation. We found that CDCA increased the cell membrane permeability by about 1.16-fold at the concentration of CDCA (40 µg/ml) as compared to the control group. However, the uptake of aminoglycosides increased about 6.68-fold compared with control. We therefore reasoned that other mechanisms might exist.”.

6. Finally, the size of some figures is too small, thus preventing from proper analysis of data presented.

Response: Thanks for the critical review and constructive advice. We have redrawn the figures to make it clear and updated the legend accordingly in the revised manuscript.

Reviewer #2 (Comments for the Author):

The authors describe the in-depth characterization of a component of from the traditional Chinese medicine Tanreqing. They have identified this agent synergizes with aminoglycosides to kill *Staphylococcus aureus*. The agent, chenodeoxycholic acid dissipates the proton motor force without disruption of the membrane. The activity against persister development initiated by aminoglycosides is compelling research for future therapeutics. The detailed work describing the physiological effects of chenodeoxycholic acid revealed the mechanism used to synergize with aminoglycosides. With the importance of the findings of this work, minor modifications to the manuscript would greatly benefit readability and the understanding of this important synergistic abilities of chenodeoxycholic acid.

1. FICI values are commonly reported by adding the FICs of both drugs ($\Sigma\text{FIC} = \text{FIC}_{\text{antibiotic}} + \text{FIC}_{\text{cdca}}$) and a value of < 0.5 is synergistic. Although the fact that these agents have synergistic activities is clear, only the FIC of the antibiotics are reported. It would be easier if the reviewer did not have to figure out the FIC of CDCA when tested with the antibiotics.

Response: Thank you for your kind suggestion. The FIC values we described in the article are all ΣFIC , which represents various antibiotics in combination with CDCA, but not $\text{FIC}_{\text{antibiotic}}$. Therefore we draw a schematic to better demonstrate the content in the FIC heat map. And all heat maps showed the $\text{FIC}_{\text{antibiotic}}$ and the FIC_{CDCA} , respectively and FICI could be calculated with the function of $\text{FICI} = \Sigma\text{FIC} = \text{FIC}_{\text{antibiotic}} + \text{FIC}_{\text{CDCA}} = (\text{MIC of antibiotic in combination}/\text{MIC antibiotic alone}) + (\text{MIC CDCA in combination}/\text{MIC CDCA alone})$.

2.The biofilm work in figure 4 shows great promise for CDCAA. However, the effect of amikacin alone or combined with CDCA is not discussed in the text for its ability to disrupt a biofilm or any combinatorial effect seen or not seen with these experiments.

Response: We thank the reviewer for pointing out this issue. We found that bacteria could survive in the biofilm even at high concentrations in AMK (green fluorescence). And we suggest that

while inhibiting the effect of the bacterial biofilm was attributed by CDCA, the red fluorescence with deepening concentration represents the effect of the drug combination.

In addition, we added Figure S2 of the separate green fluorescence (live bacteria) channels to the supplementary data to support this conclusion.

Fig. S2. CDCAA is effective against MRSA biofilm.

Representative separate green fluorescence (alive bacteria) channels Z-axis (25 μm) overlay image of MRSA ATCC 43300 biofilms (triplicates) treated with amikacin alone (top) or in combination with CDCA (bottom). The scale bar is 70 μm . The fluorescence contrast values of each image was set at the same condition with a minimum of 500 and maximum of 2000 a.u. (arbitrary unit).

Minor comments:

1. TCM in the abstract is not mentioned what it stands for.

Response: Thanks for the critical review and we have made the revisions in the manuscript.

2. There are a lot of acronyms throughout the text. Perhaps remove those that are used less than 5 times in the article to make it easier on the reader. (SCV, EMA, NMPA, FDA, etc.)

Response: Thanks for the critical review and we have made the revisions in the manuscript by removing EMA, NMPA, FDA, and PI, while we keep SCV since it has been used throughout the text for more than 5 times.

3. WHO on line 61 does not say what it stands for

Response: Thanks for the critical review. We have made the revisions in the manuscript.

4. Supplemental table 1 has bolded values and it is not clear or stated why they are bolded

Response: We thank the reviewer for noticing the details. We aimed to highlight strains ineffective (resistant) to FA treatment in bold. This change has been added in the table note.

4. Line 207 “efficacy or CDCAA efficacy” should be reworded

Response: Thanks for the critical review. As requested, the first sentence in line 207 was revised.

5. Line 221 Cell line HEI-OCI is not described well enough. This is an uncommon cell line and it

should be acknowledged as a mammalian cell line and provided with the reason for this cell line choice over the commonly used kidney and liver cell lines.

Response: Thanks for the critical review. It has long been known that the major irreversible toxicity of aminoglycosides is ototoxicity. Among them, streptomycin and gentamicin are primarily vestibulotoxic, whereas amikacin, neomycin, dihydrostreptomycin, and kanamicin are primarily cochleotoxic. HEI-OCI (House Ear Institute-Organ of Corti 1) cells, a type of mouse cochlear hair cell, have been widely used as an *in vitro* system to study cellular and molecular mechanisms related to hearing loss, such as cell death, apoptosis, survival, proliferation, senescence and autophagy, oxidative stress, etc (DOI: 10.1080/15548627.2017.1359449). Therefore, we chose to verify the ototoxicity of the combination with HEI-OCI.

6. In the discussion G⁺ and G⁻ are abbreviated but not anywhere else in the text. It starts in the last section of the results. Please ensure uniformity throughout the text.

Response: Thanks for the comments. We have thoroughly changed “Gram-negative” and “Gram-positive” to “G⁻” and “G⁺”, respectively, in the revised version.

7. Line 390 italicize *S. aureus*.

Response: Thanks for the comments. As requested, *S. aureus* in line 390 was revised.

8. The statistical analysis used to find the P values should also be included in the figure legends so the reader can see what equation was used for those values. And there is no mention of corrections used for multiple analyses. The exception to this was Fig 5B that included all information.

Response: Thanks for the critical comments. We supplement the corresponding instructions in all the graph notes used for the statistical analysis.

November 15, 2022

Dr. Qing Wei
Nanchang Institute of Technology
Yingxiong Road
Nanchang, Jiangxi 330044
China

Re: Spectrum02430-22R1 (Chenodeoxycholic acid and amikacin combination enhance eradication of *Staphylococcus aureus*)

Dear Dr. Qing Wei:

Thank you for submitting your manuscript to Microbiology Spectrum. It has now been reviewed by one expert who considers that the work has substantially improved, but there are still a number of suggestions for improvement. The comments are included at the bottom of this letter. In order for your paper to become acceptable for publication, it must be carefully revised in such a way that it satisfactorily addresses all issues raised by the reviewer. Please, pay especial attention to concerns regarding Figure two. In case you have only used the MSSA strain, the conclusion of CDCA+amikacin killing MRSA should be eliminated. In case you have actually used MSSA and MRSA, an explanation about why the MRSA strain has been tested with such a low amikacin concentration should be given and discussed in the manuscript. When submitting the revised version of your paper, please provide (1) point-by-point responses to the issues raised by the reviewers as file type "Response to Reviewers," not in your cover letter, and (2) a PDF file that indicates the changes from the original submission (by highlighting or underlining the changes) as file type "Marked Up Manuscript - For Review Only". Please use this link to submit your revised manuscript - we strongly recommend that you submit your paper within the next 60 days or reach out to me. Detailed instructions on submitting your revised paper are below.

Link Not Available

Sincerely,

Cristina Solano

Journals Department
Reviewer comments:

Reviewer #1 (Comments for the Author):

The authors have improved their manuscript in comparison with the original submission, although there are still a few points that need further work, mainly related with results and figures:

- 1) Figure 2D is clear: the main text says that this corresponds to MSSA, the figure legend says MSSA ATCC 25923, and experiments have been done with 0.5 µg/ml amikacin, which is 0.5 x MIC of the MSSA strain. However, Figure 2E is really unclear, because the main text says that this corresponds to MRSA, the figure legend says MRSA ATCC 25923 (please note that this is the ATCC number for the MSSA strain; that for the MRSA strain is 43300), and experiments have been done with 10 µg/ml amikacin, which is 10 x MIC of the MSSA strain (why? 10 µg/ml would correspond to 0.15 x MIC of the MRSA strain). So, it looks as if both figures have done with the SAME strain (the MSSA strain only) but using two different concentrations of amikacin (0.5 and 10 µg/ml, being respectively 0.5 and 10 times the MIC of the MSSA strain). If so, then the text should be revised because references to antimicrobial activity of amikacin+CDCA against the MRSA strain should be changed to MSSA strain (for example, in lines 33, 152, 155, 156, 161, 163, 187, and probably others along the text).
- 2) Figure 4B, the scale bar is not shown
- 3) Figure 4D, what is the meaning of 'normal cells'? does it mean normal size? How has been determined? How many cells were inspected?
- 4) Legend to figure 5A, the sentence 'Model group means' is uncomplete
- 5) Figure 5, it is difficult to observe the black triangle. Please, try to increase contrast by adding a white line, using another color, or any other way.
- 6) Figure 6G, legend and figure do not correspond, legend only explains partially the content of the figure.
- 7) Figure 7A, change 'promoted' to increased or up-regulated, and 'repressed' to decreased or down-regulated

Staff Comments:

Preparing Revision Guidelines

Please return the manuscript within 60 days; if you cannot complete the modification within this time period, please contact me. If you do not wish to modify the manuscript and prefer to submit it to another journal, please notify me of your decision immediately so that the manuscript may be formally withdrawn from consideration by Microbiology Spectrum.

Reviewer comments:

Reviewer #1 (Comments for the Author):

The authors have improved their manuscript in comparison with the original submission, although there are still a few points that need further work, mainly related with results and figures:

1) Figure 2D is clear: the main text says that this corresponds to MSSA, the figure legend says MSSA ATCC 25923, and experiments have been done with 0.5 µg/ml amikacin, which is 0.5 x MIC of the MSSA strain. However, Figure 2E is really unclear, because the main text says that this corresponds to MRSA, the figure legend says MRSA ATCC 25923 (please note that this is the ATCC number for the MSSA strain; that for the MRSA strain is 43300), and experiments have been done with 10 µg/ml amikacin, which is 10 x MIC of the MSSA strain (why? 10 µg/ml would correspond to 0.15 x MIC of the MRSA strain). So, it looks as if both figures have done with the SAME strain (the MSSA strain only) but using two different concentrations of amikacin (0.5 and 10 µg/ml, being respectively 0.5 and 10 times the MIC of the MSSA strain). If so, then the text should be revised because references to antimicrobial activity of amikacin+CDCA against the MRSA strain should be changed to MSSA strain (for example, in lines 33, 152, 155, 156, 161, 163, 187, and probably others along the text).

Response: We thank the reviewer for pointing out this issue. We have changed the strain name in Figure 2E from “MSSA ATCC 25923” to “MRSA ATCC 43300” and corrected the MIC. To be sure of this, we thoroughly examined the “MSSA” and “MRSA” throughout the paper and are sure that the descriptions of strains in the main text are correct.

2) Figure 4B, the scale bar is not shown

Response: Thanks for the critical review. The color of the scale bar in Figure 4B is too light to see and we have added the new scale.

Revised Fig. 4

3) Figure 4D, what is the meaning of 'normal cells'? does it mean normal size? How has been determined? How many cells were inspected?

Response: Thanks for the comment. We have changed the wording from “ normal cells” to “normal size cells”. We enumerated CFU separately at each time point and observed colony morphology, and cells with a diameter less than half of the normal size cells (1mm) were also identified as SCV. In addition, at each time point in each group, there were three biological replicates and three technical replicates. Therefore, each group should have 270~2,700 cells, and

in total about 1,000 cells were counted for each group.

4) Legend to figure 5A, the sentence 'Model group means' is incomplete

Response: We thank the reviewer for pointing out this issue. We accidentally added this sentence in the last modification, which have been deleted.

5) Figure 5, it is difficult to observe the black triangle. Please, try to increase contrast by adding a white line, using another color, or any other way.

Response: Thanks for the critical review and constructive advice. We have updated Figure 5 and made the black triangle more visible by adding a white border.

Revised Fig. 5

6) Figure 6G, legend and figure do not correspond, legend only explains partially the content of the figure.

Response: Thanks for the critical review and constructive advice. We apologize for the lack of detail and have improved the description in the legend of Figure 6 as follows:

“ATP levels of MRSA ATCC 43300 were measured after treatment with amikacin together with or without different concentrations of CDCA for 6 h.” In addition, control groups were treated with amikacin alone.

7) Figure 7A, change 'promoted' to increased or up-regulated, and 'repressed' to decreased or down-regulated.

Response: Thanks for the critical review and constructive advice. We have updated Figure 7A and change “promoted” to “increased”, and “repressed” to “decreased”.

Revised Fig. 7

December 7, 2022

Dr. Qing Wei
Nanchang Institute of Technology
Yingxiong Road
Nanchang, Jiangxi 330044
China

Re: Spectrum02430-22R2 (Chenodeoxycholic acid and amikacin combination enhance eradication of *Staphylococcus aureus*)

Dear Dr. Qing Wei:

Your manuscript has been accepted, and I am forwarding it to the ASM Journals Department for publication. You will be notified when your proofs are ready to be viewed.

Sincerely,

Cristina Solano
Editor, Microbiology Spectrum
